# The Association between Molecular Initiating Events and Drug-Induced Hiccups

**DOI:** 10.3390/ph17030379

**Published:** 2024-03-16

**Authors:** Ryuichiro Hosoya, Reiko Ishii-Nozawa, Tomoko Terajima, Hajime Kagaya, Yoshihiro Uesawa

**Affiliations:** 1Department of Medical Molecular Informatics, Meiji Pharmaceutical University, Tokyo 204-8588, Japan; 2Laboratory of Clinical Pharmaceutics, Department of Clinical Pharmacy, Faculty of Pharmaceutical Sciences, Shonan University of Medical Sciences, Yokohama 244-0806, Japan; tomoko.terajima@sums.ac.jp (T.T.); hajime.kagaya@sums.ac.jp (H.K.); 3Department of Pharmacology, Meiji Pharmaceutical University, Tokyo 204-8588, Japan; reiko-in@my-pharm.ac.jp

**Keywords:** drug-induced hiccups, FAERS, nuclear receptor, stress response pathways

## Abstract

Hiccups can significantly reduce the quality of life of patients and can occur as a drug side effect. Previous reports have revealed sex-specific differences in the incidence of drug-induced hiccups. However, the pathogenesis of drug-induced hiccups remains unknown, and there is limited evidence on its treatment or prevention. This study examined molecular initiating events (MIEs), which are the starting point of adverse events, to investigate the drug-induced pathways of hiccups. We extracted drugs suspected to cause hiccups using the FDA Adverse Event Reporting System, a large database on adverse drug reactions. Information on drugs suspected to be associated with hiccups was extracted from the overall population and sex-specific subgroups were divided. In each data table, the predicted activity values of nuclear receptors and stress response pathways for each drug were calculated using the Toxicity Predictor, a machine-learning model. Transforming growth factor-beta and antioxidant response elements were considered an independent factor for hiccups in the male and female subgroups, respectively. This report first examined one of the mechanisms of drug-induced hiccups and identified MIEs associated with drug-induced hiccups. The use of an adverse event database and the machine-learning model, Toxicity Predictor, may be useful for generating hypotheses for other adverse effects with unknown mechanisms.

## 1. Introduction

Hiccups are a common symptom experienced by many people and are mainly caused by myoclonus of the diaphragm [1]. Hiccups have been associated with speech, sleep, and swallowing disorders, weight loss, fatigue, and insomnia [2]. Although not life-threatening, hiccups can significantly reduce the quality of life of patients. If hiccups occur as a drug side effect, they can also impact the choice of drug treatment. Therefore, controlling hiccups that are drug side effects is important for drug therapy.

Hiccups are brief, involuntary spasms of the muscles of the diaphragm accompanied by coordinated contractions of the glottis closure muscle group [1]. The afferent and efferent pathways of the hiccup reflex arch involve the glossopharyngeal nerve (nineth cranial nerve), vagus nerve (tenth cranial nerve), nucleus of the solitary bundle tract, nucleus ambiguous, and peroneal nerve [3]. The exact mechanism of the hiccup reflex arch is unclear. However, neurotransmitters such as gamma-aminobutyric acid (GABA), serotonin, and dopamine have been associated with the onset of hiccups. Reflex arches are mediated by central neurotransmitters (GABA, dopamine, and serotonin) and peripheral neurotransmitters (epinephrine, norepinephrine, acetylcholine, and histamine) [4,5]. There are no reports on sex differences or physical information about common hiccups experienced by healthy individuals. However, male patients have a high incidence of intractable or persistent hiccups [6].

Several reports have described drug-induced hiccups. Lee et al. revealed a male predominance for peripheral hiccups based on a meta-analysis. However, there were no sex differences in hiccups caused by central nervous system disorders [6]. In a study about the risk factors of patient-induced hiccups, low body mass index, nausea and vomiting, and cancer drug therapy were associated with hiccups [7]. In other reports, cisplatin [8,9] and dexamethasone [10] have been suspected to affect the development of hiccups. However, no studies have analyzed the risk factors for drug-induced hiccups using data from large databases. Therefore, in a previous study, drug and patient information related to hiccups was analyzed using data from the Japanese Adverse Drug Event Report (JADER) database and the FDA Adverse Event Reporting System (FAERS) database via data mining [11,12]. Dexamethasone and several anticancer drugs were identified as independent risk factors for hiccups. We also established a visualization method for suspected drugs in the adverse drug reaction database. In a prior study using data from the JADER database, data on the drugs suspected to affect hiccups were extracted, and the results showed that a patient’s height is a risk factor for hiccups [11]. Furthermore, in an analysis using FAERS data stratified according to sex, the drugs suspected to affect hiccups differed significantly between male and female patients. Among them, nicotine was commonly suspected to be associated with hiccups in male and female individuals [12]. Hiccups have male predominance [6]; however, their cause has not been elucidated. No studies have yet validated whether the male predominance of hiccup occurrence is attributed to drug sensitivity or neurological sex differences.

As shown in a previous study, several case reports and database studies have assessed the association between hiccups and drugs. However, to the best of our knowledge, only a few studies have investigated the mechanism of drug-induced hiccups. The small number of cases involving drug-induced hiccups and the difficulty in predicting their onset are some reasons for the lack of research. To treat and prevent drug-induced hiccups, the causative mechanisms should be identified. Therefore, the molecular chemistry related to the induction of hiccups based on the suspected drug information identified using the FAERS database was used. By handling information on suspected drugs obtained from several sources, rare cases of adverse drug reactions, such as hiccups, can be evaluated. These results may provide leads for novel research to elucidate the pathogenic mechanism of hiccups.

Numerous factors are involved in the mechanism of the effects of adverse drug reactions. From the perspective of toxicology, molecular initiating events (MIEs) in the adverse outcome pathway (AOP) represent an important concept when considering the mechanism of drug-induced adverse effects [13]. MIEs denote the first interaction between a molecule and a biomolecule or the biological system in the AOP. Their targets include nuclear receptors (NRs) and stress response pathways (SPs). NRs are intracellular proteins that regulate DNA transcription in the cell nucleus by binding to hormones and other substances. NRs bind directly to DNA and regulate gene expression, and they are associated with the development, homeostasis, and metabolism of the organism. If a ligand binds to a NR, a conformational change occurs, which activates the receptor and subsequently regulates gene expression [14]. SPs are biochemical processes that regulate the response to various stress signals in cells. Endocrine-disrupting chemicals disrupt the endocrine system by interacting with NRs and SPs, causing various adverse developmental, reproductive, neural, and immunological effects in humans and wildlife [15]. Moreover, NRs and SPs may be involved in adverse drug events.

Drug-induced hiccups may develop hours or days after drug administration and persist for an extended period [7,10]. Therefore, these phenomena are not controlled on a millisecond-by-millisecond basis but rather through the development of several molecular pathways from the time of drug administration. Considering these facts, we believe that searching for NRs/SRs as MIEs involved in the induction of drug-induced hiccups will help elucidate the underlying mechanisms. Therefore, this study aimed to identify NRs and SPs associated with drug-induced hiccups using Toxicity Predictor, a machine-learning model. We aimed to construct a combined database to generate hypotheses for clinical questions. We also aimed to establish our methodology so that this method can be applied to other side effects with unknown mechanisms.

## 2. Results

### 2.1. Prediction of MIEs Associated with Hiccups

There were 26,316 reports of hiccup reversals in the analysis database. Crosstabulation tables were created for each drug, and the *p*-values for Fisher’s exact test and the reported odds ratio (ROR) were calculated. Figure 1 presents a scatter plot (volcano plot) depicting the results of the univariate analysis for all patient data. There were 169 drugs with >1000 reports and those considered significant (*p* ≤ 0.001 based on Fisher’s exact test). Appendix A shows a list of 156 drugs, including the ATC names and classifications of the suspected drugs. This file only lists important suspect drugs and drugs classified by the ATC system. Therefore, there is a slight difference in the number of drugs used in the analysis. Univariate analysis using lnROR as the binary classification value for the objective variable revealed that the significant factors were the estrogen receptor alpha with antagonist (ERα), progesterone receptor (PR), androgen receptor with antagonist (ARα), transforming growth factor-beta (TGF-β), and sonic hedgehog (Shh). Table 1 depicts the activity types and the results of both univariate and multivariate analyses for each MIE. The results of the nominal logistic regression analysis were submitted to the MIE. The results of the univariate analysis with the presence or absence of hiccups as the objective variable were significant. In the multivariate analysis, odds ratios were used to confirm independent factors related to hiccups.

### 2.2. Prediction of MIEs Associated with Hiccups in Male Patients

There were 18,663 reports of hiccup reversals in the data table for men. Crosstabulation tables were created for each drug, and the *p*-values for Fisher’s exact test and the ROR were calculated. Figure 2 presents a volcano plot of the key suspect drugs for hiccups in male patients. There were 122 drugs with >1000 reports and those considered significant (*p* ≤ 0.001 based on Fisher’s exact test) in men. Appendix A depicts a list of 108 drugs, including the ATC names and classifications of the suspected drugs. This file only lists important suspect drugs and drugs classified by the ATC system. Therefore, there is a slight difference in the number of drugs used in the analysis. Univariate analysis using lnROR as the binary classification value for the objective variable revealed that PR, constitutive androstane receptor (CAR), Shh, androgen receptor lbd (ARlbd), and TGF-β were identified as significant factors. Table 2 presents the activity types and the results of both univariate and multivariate analyses for each MIE. The results of the nominal logistic regression analysis were submitted to the MIE, which showed significant results for the univariate analysis with the presence or absence of hiccups as the objective variable. In the multivariate analysis, odds ratios were used to confirm independent factors related to hiccups in male patients.

### 2.3. Prediction of MIEs Associated with Hiccups in Female Patients

There were 6268 reports of hiccup reversals in the data table for women. Crosstabulation tables were created for each drug, and the *p*-values for Fisher’s exact test and the ROR were calculated. Figure 3 presents a volcano plot of the key suspect drugs for hiccups in female patients. There were 45 drugs with >1000 reports and those considered significant (*p* ≤ 0.001 based on Fisher’s exact test) in female participants. Appendix A shows a list of 41 drugs, including the ATC names and classifications of the suspected drugs. This file only lists important suspect drugs and drugs classified by the ATC system. Therefore, there is a slight difference in the number of drugs used in the analysis. Based on the univariate analysis using lnROR as the binary classification value for the objective variable, the significant factors were antioxidant response element (ARE), CAR, and ARa. Table 3 depicts the activity types and the results of both univariate and multivariate analyses for each MIE. The results of the nominal logistic regression analysis were submitted to the MIE, which showed significant results for the univariate analysis with the presence or absence of hiccups as the objective variable. In the multivariate analysis, odds ratios were used to confirm independent factors related to hiccups in female patients.

Figure 4 presents the results of the univariate analysis for the onset of hiccups and activity of each MIE during drug treatment. By taking the negative value of the ordinary logarithm of Fisher’s exact test on the vertical axis and the natural logarithm of the odds ratio on the horizontal axis to create a scatter plot, the MIEs likely to be strongly associated with the development of hiccups are indicated at the top. We visually confirmed that the associated MIEs differed for men and women.

## 3. Discussion

This study used data from the FAERS database, an adverse drug reaction database, to identify the key suspected drugs causing hiccups. Furthermore, the association between the suspected drugs and MIEs was investigated. To the best of our knowledge, there is limited information on the mechanism of drug-induced hiccups, and no reports based on adverse events reporting systems have revealed the mechanism of drug-induced hiccups. In addition, this is the first study that aimed to examine the MIEs, especially NRs/SRs, of drug-induced hiccups using Toxicity Predictor and the FAERS database. The Toxicity Predictor can predict the agonist and antagonist activities of a drug in the MIE, which is the starting point of action of an adverse effect. Previous reports have described methods for evaluating the MIEs of adverse effects using Toxicity Predictor [13]. We believe that such methods are reliable [16].

To investigate the association between each suspected drug and NRs/SPs, the objective variable was a binary variable of lnROR > 0 (positive signal) or <0 (negative signal). A method to identify prophylactic and therapeutic agents against side effects by focusing on drugs with positive and negative signals is a method of drug repositioning and has been attracting significant attention in recent years [17,18,19]. This study examined the involvement of NRs/SPs in hiccups by comparing the MIE activities of drugs with positive and negative signals for drug-induced hiccups.

The data used in this study were classified into the overall cohort and the male and female subgroups. Drug-induced hiccups occur more frequently in men [6]. In our previous study using the adverse drug reaction database, hiccups were also more common in men, and the distinctions regarding the drugs suspected were noted between male and female patients [11,12]. Based on these findings, our objective was to identify the differences in NRs and SPs involved in the development of hiccups between male and female patients. The multivariate analysis identified the TGF-β agonist as an independent factor associated with hiccups in the overall dataset. In the male subgroup, TGF-β was an independent factor based on the univariate and multivariate analyses. In the female group, ARE was an independent factor in the univariate and multivariate analyses.

To the best of our knowledge, this report is the first to explore the association between drug-induced hiccups and TGF-β. Toxicity Predictor, a toxicity prediction program, centers its predictions on the toxicity results of Tox21. To assess the cytotoxicity of a compound in Tox21, the transduction process by which TGF-β binds to the TGF-β receptor and acts on nuclear gene expression via the Smad protein is being monitored.

TGF-β is an important cytokine that maintains bodily homeostasis. It has been detected in a broad array of tissues and organs and is known for its roles in inhibiting cell proliferation and inducing cell differentiation and apoptosis in various cell types. In addition, TGF-β is involved in processes such as cell differentiation, migration, and adhesion. Moreover, it plays significant roles in diverse mechanisms including ontogeny, tissue remodeling, wound healing, inflammation, immunity, and cancer invasion and metastasis. TGF-β was identified in the cerebral spinal fluid of mice exhibiting exhaustion. Thus, its use as a fatigue marker has gained attention [20].

TGF-β, which is ubiquitous and produced in several tissues and cells, is released in an inactive, latent form (latent TGF-β) that cannot bind to receptors. It becomes activated when in the vicinity of target cells where it transforms into its active form (active TGF-β) that can bind to receptors and exert its effects. Astrocytes are known to be one of the major cells producing and releasing TGF-β in the brain where GABA receptors are located, with drug action and viral infection having been reported as facilitating factors [21,22]. Known external factors that activate TGF-β include acids, alkalis, heat, reactive oxygen species, vitamin A, vitamin D, anti-estrogens, bleomycin, and dexamethasone [23]. The TGF-β family receptors have been categorized into the following three types: type I, type II, and type III. Within the signaling pathway, ligand-bound type II receptors initiate the activation of type I receptors via phosphorylation, followed by autophosphorylation. This process activates and binds Smad2 and Smad3. After phosphorylation, Smad2/3 associates with Smad4, leading to the translocation of this complex into the nucleus where it functions as a transcription factor (Figure 5). The TGF-β/Smad signaling pathway is important for regulating cell development and growth. Disruptions in this pathway have also been significantly associated with tumor development [24].

In this study, TGF-β was identified as a novel factor implicated in the induction of hiccups. Previous research has shown that drug-induced hiccups are more prevalent in men and that there are sex differences in the drugs commonly implicated [12]. Based on these findings, TGF-β was identified as an associated factor in both the overall dataset and specifically in the male subgroup. This discovery could significantly enhance our understanding of sex differences in hiccups. Moreover, hiccups are associated with GABA, an inhibitory system neuron, with the GABA derivative baclofen used for treating hiccups [25]. TGF-β promotes the growth of dopamine nerves and regulates the GABAergic nervous system. Although the exact intracellular mechanism of TGF-β is not completely understood, some studies have revealed that Erk1/2 and GSK3β might increase GABAergic neurotransmission by inhibiting the phosphorylation of gephyrin, a scaffolding protein for the GABA receptor [26]. Hence, TGF-β can be associated with the onset of hiccups via its influence on the GABAergic nervous system. Current evidence showing an association between TGF-β and drug-induced hiccups is limited. Hence, it is considered a possible MIE in the mechanistic understanding of hiccups. Ongoing research about the TGF-β signaling pathway can further clarify its role in neurotransmission.

In the analysis of the female subgroup, ARE, one of the mechanisms associated with protection against oxidative stress, was found to be a factor associated with hiccups. The odds ratio (OR: 0.08 [0.01–0.70]) indicates a negative signal for drug-induced hiccups, which may be associated with hiccup inhibition. The Tox21 program monitors ARE activation via the Nrf2/antioxidant response signaling pathway to assess compound toxicity.

Nrf2-ARE is an important signaling pathway that regulates the expression of antioxidant enzymes. In the cytosol, Nrf2 binds to Kelch-like ECH-associated protein 1 (KEAP1) in the cytoplasm, and its activity remains low under normal physiological conditions [27]. When cells are exposed to oxidative stress, Nrf2 unbinds KEAP1 and moves into the nucleus to bind with ARE (Figure 6). This promotes the transcription and expression of a series of antioxidant enzymes, including heme oxygenase-1 (HO-1), superoxide dismutase (SOD), and thioredoxin (Trx) [28].

Currently, there are no reports on hiccups and oxidative stress. However, drugs that have been reported as important suspect drugs for hiccups (e.g., nicotine, dexamethasone, and anticancer drugs) increase oxidative stress. Nicotine causes increased oxidative stress, leading to greater neuronal apoptosis, DNA damage, reactive oxygen species, and lipid peroxides. Nicotinic acetylcholine receptors (nAChRs) have been identified in tissues other than those in the nervous system, and their effects on nicotinic receptors have been associated with acute and chronic effects [29]. Glucocorticoid levels increase if the organism is stressed, which is accompanied by an increase in free radicals. Dexamethasone ingestion mimics the adverse effects of increased corticosterone in vivo [30]. These findings may explain why hiccups are associated with oxidative stress. The fact that the current study showed the involvement of ARE in the inhibitory side of hiccups may also support an association between hiccups and oxidative stress. We believe that ARE was an independent factor only in the female subgroup because of sex differences regarding the suspected drug for hiccups. Hence, future studies on oxidative stress and sex differences in antioxidant mechanisms should be performed. Regarding the independence of factors in this subgroup analysis, the results of the multiple logistic regression analysis showed that ARE had a *p*-value of <0.05 based on the likelihood ratio test but not the Wald test (Table 3). As a possible cause for the uncertainty of the results, the small number of female patients used in this analysis might have affected the test. It is necessary to wait for the accumulation cases involving female patients to corroborate the current results and ensure the validity of the discussion.

The results of the current study indicated that should pathways related to NRs and SPs be associated with the induction of hiccups, the mechanism of their development may differ between men and women (Figure 4). However, given the clear sex differences in drug-induced hiccups, some sex hormone receptor involvement in the mechanism underlying the onset of drug-induced hiccups can be expected. The NR activities we used to predict hiccups included the agonist and antagonist activities of the androgen and estrogen receptors. However, no significant correlation had been observed between these activities and the induction of hiccups. The limited number of reports on hiccups compared with other side effects may explain why no relationship between sex hormone receptors and hiccups could be detected.

Another study on TGF-β and sex differences observed a difference in gene expression in the lens of male and female rats with cataract formations induced by TGF-β [31]. The mentioned report suggests that sex differences may be associated with the expression and activation of TGF-β. Given that reports on sex differences in TGF-β in humans are limited, future studies need to examine these sex differences in more detail.

## 4. Materials and Methods

### 4.1. Database Information

This study used data from the FAERS database. The FAERS is a large database of information on adverse drug events collected from all regions globally. In total, 14,836,467 cases reported between 1 January 2004 and 31 March 2022 were downloaded from the FDA website [32]. Figure 7 shows the procedure for creating the data tables.

### 4.2. Definitions of Adverse Events and Suspected Drugs

The International Council for Harmonization of Technical Requirements for Pharmaceuticals for Human Use developed MedDRA [33] as a repository of medical terminology for symptoms, signs, and diseases. In this study, hiccups were treated as a common term for adverse drug reactions. According to the adverse drug reaction reported, the drugs listed in the FAERS database are classified as primary, secondary, concomitant, and interacting drugs. In this study, all drugs were treated as suspected drugs.

### 4.3. Extraction of Drugs Suspected of Causing Hiccups

The drugs suspected to induce hiccups vary widely according to sex [11,12]. Thus, the overall analysis data table and subgroup tables divided based on sex were used to extract the suspected drugs (Figure 7). For the subgroup analysis, cases of unknown gender were excluded. The number of “hiccups” reported as adverse effects for each drug was calculated as the presence of hiccups. In each group, cross-tabulations were performed based on two categories (the presence/absence of hiccups and the presence/absence of the suspect drug) to calculate the ROR, and any significant result was evaluated using Fisher’s exact test (Table 4). Appendix A shows the *p*-values, ROR, and number of reports for each drug. A scatterplot (volcano plot) was then created by plotting the negative log of the *p*-value from Fisher’s exact test for each group on the *y*-axis and the natural log of the ROR (lnROR) on the *x*-axis. The drugs with a *p*-value of <0.001 based on Fisher’s exact test and with an ROR of >1 were suspected to cause hiccups. Appendix A shows the list of important suspected drugs and their ATC classification, which were divided according to the overall population and sex-specific subgroups.
$$\mathrm{ROR}=\frac{n_{11}/n_{12}}{n_{21}/n_{22}}=\frac{n_{11}\cdot n_{22}}{n_{12}\cdot n_{21}}$$

### 4.4. MIE Activity Prediction Using Toxicity Predictor

Toxicity Predictor [34] is a web-based application developed and operated by Meiji Pharmaceutical University. In this study, it was used to explore the AOPs of drug-induced hiccups. Furthermore, Toxicity Predictor, which was developed as part of the Drug Discovery Information System Development Project of the Japan Agency for Medical Research and Development, can convert molecular structures obtained from input files into three-dimensional (3D) structures. In addition, it is a quantitative structure–activity relationship (QSAR) model-based toxicity prediction system that can evaluate agonist and antagonist activity against 56 MIEs. QSAR is a machine-learning algorithm that can learn mathematical associations between the chemical structure of a molecule and its biological or chemical activity to create a predictive model. MIE 3D structures and predicted results can be downloaded in the SDF and CSV formats, respectively [16].

To predict MIE activity using Toxicity Predictor, the simplified molecular-input line-entry system was combined for each drug in each table. Using the Toxicity Predictor, the agonist and antagonist activity (MIE activity) of drugs targeting NRs and SPs for each group was calculated. The cutoff values for the predicted MIE activity were determined using the Youden method. The calculated MIE activities were normalized to establish a cutoff value of 0.5. Thus, the predicted labels of compounds with normalized predicted values of >0.5 were assigned a value of 1. Meanwhile, those with predicted values of <0.5 were assigned a value of 0. Appendix A depicts the predicted values.

### 4.5. MIEs Associated with Hiccups

In each data table for the overall population and for the male and female subgroups, only the drugs suspected to be associated with hiccups were extracted. The selection criteria included the presence of at least 1000 adverse event reports and a *p*-value of ≤ 0.05, as determined using Fisher’s exact test. The objective variable was a binary variable with an lnROR of >0 or lnROR of <0. Univariate analysis was performed using the MIE activity (1-0) of each drug as the explanatory variable. In addition, to extract independent risk factors for each factor, logistic regression analysis was performed with lnROR as the objective variable and NRs and SPs, which were significant in the univariate analysis, as explanatory variables.

### 4.6. Statistical Analysis

In this study, the objective variable was the binary classification of the ROR (lnROR ≥ 0, lnROR < 0). Hence, the ROR of a drug is significantly higher for a given adverse effect. For example, the adverse event is more likely to occur in other drugs. The explanatory variables were the probability for MIE activity calculated using Toxicity Predictor, normalized to a cutoff value of 0.5, and binary classified values. Univariate analysis was conducted to evaluate the NRs and SPs associated with hiccups. If the result of Fisher’s exact test was significant in the positive direction, it was considered a factor associated with the induction of hiccups. In contrast, if the result of Fisher’s exact test was significant in the negative direction, it was a factor related to the inhibition of hiccup induction. Using MIEs that were significant in the univariate analysis, a multivariate analysis was performed to extract independent risk factors for hiccup inversions. The likelihood ratio test and the Wald test were used to evaluate the independence of each factor. The significance level was set at *p* < 0.05. The pairwise method was used to evaluate internal correlation. An internal correlation was considered to exist if the Spearman’s rank-order correlation coefficient [ρ2] was >0.9. In this study, no internal correlation was found, and each factor was treated as independent. All analyses were performed using JMP Pro 17 software (SAS Institute Inc., Cary, NC, USA).

### 4.7. Limitation

This study is entirely database based. Therefore, it has several limitations. The NR activity values used in this study were predicted using the QSAR and may differ from the activity values obtained experimentally. To compare the actual and predicted values, drugs with >5000 reported cases and ROR > 1 were selected among the key suspect drugs. For TGF-β and ARE, the presence or absence of activity by actual values (Positive (1)/Negative (0)) and the presence or absence of activity by predicted values were compared (Appendix A). For ARE, only 2 of the 14 drugs available for comparison had different results. For TGF-β, all results were the same. Although these findings suggest that the accuracy of the predictions is reliable, it should be noted that some results differ and are based on predicted values.

The FAERS, the other database used in this study, which comprises a spontaneous report of adverse drug reactions, is particularly useful for detecting rare and serious adverse drug reactions. Moreover, it is an important source of information for evaluating the safety of drugs. In particular, the FAERS database has a stronger reporting bias because it includes cases in which patients who have taken the drug are enrolled as reporters [35]. In addition, the presence of drugs that have already been found to be associated with adverse events may affect the occurrence of adverse drug event-associated signals [36].

In this study, the number of reports and the *p*-value obtained using the exact test were examined along with the ROR obtained via univariate analysis when detecting the signal, thereby avoiding a simple comparison of RORs and treating them semiquantitatively. This strategy was based on the hypothesis that a signal detection indicator considered highly significant based on the number of reports and *p*-values would have excellent reliability [12]. Additionally, the number of adverse drug reaction reports varies depending on when the drug was launched. If the number of drug reports is low, the OR may be unreliable. To minimize the impact of the number of drug reports available, our analysis only included situations with >1000 reports of adverse drug reactions. Evaluation of drugs new to the market should be performed with caution until more reports become available.

## 5. Conclusions

The FAERS database was used to examine the association between NRs and SPs as well as drug-induced hiccups. We successfully extracted MIEs associated with drug-induced hiccups. Current case reports have reported on hiccups. However, to the best of our knowledge, this report is the first to consider the pathogenesis of hiccups. This method can be applied to other adverse drug reactions, and it may be useful for elucidating the mechanism of adverse drug events.

## Figures and Tables

**Figure 1 pharmaceuticals-17-00379-f001:**
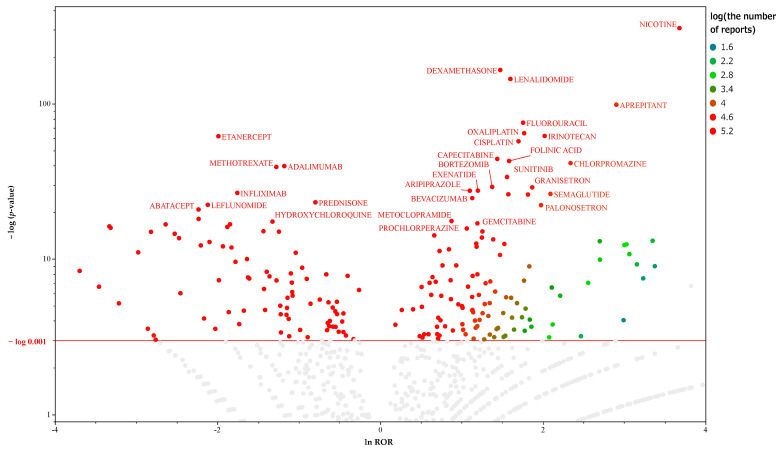
Drugs associated with hiccups. This volcano plot was created by plotting the negative logarithm of the *p*-value (−log_10_*p*) based on Fisher’s exact test on the *y*-axis and the natural logarithm of the ROR (lnROR) on the *x*-axis. In all patients, drugs with >1000 reports and those considered as significant (*p* ≤ 0.001 based on Fisher’s exact test) were included in the volcano plot. The color of the individual points represents differences in the log of the number of reports for each drug. The gray plots indicate drugs with a *p*-value of >0.001 or <1000 reported cases. The red line on the *y*-axis represents a *p* = 0.001. In this scatter plot, the signal is larger for the points (drugs) plotted in the upper right corner. The blue-to-red colors represent the number of times an adverse drug reaction was reported.

**Figure 2 pharmaceuticals-17-00379-f002:**
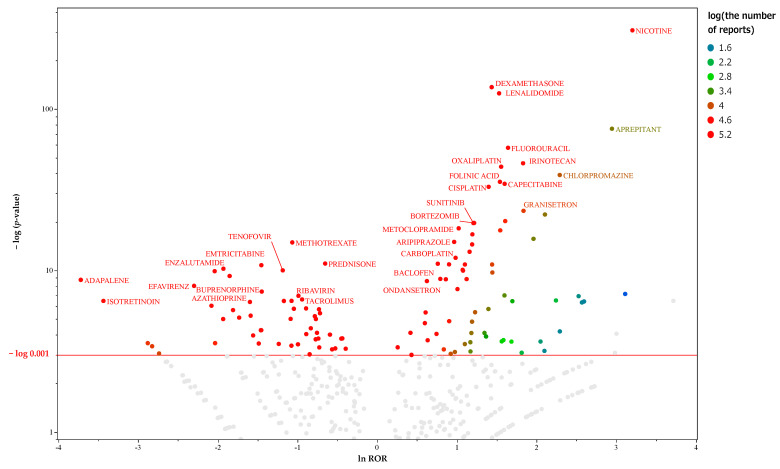
Drugs associated with hiccups in male participants. This volcano plot was created by plotting the negative logarithm of the *p*-value (−log_10_*p*) based on Fisher’s exact test on the *y*-axis and the natural logarithm of the ROR (lnROR) on the *x*-axis. The red line on the *y*-axis represents *p* = 0.001. In male patients, drugs with >1000 reports and those that are significant (*p* ≤ 0.001 based on Fisher’s exact test) are included in the volcano plot. The color of the individual points represents differences in the log of the number of reports for each drug. The gray plots indicate drugs with *p* > 0.001 or <1000 reported cases. In this scatter plot, the signal is larger for the points (drugs) plotted in the upper right corner. The line on the *y*-axis represents the total average. The blue-to-red colors represent the number of times an adverse drug reaction was reported.

**Figure 3 pharmaceuticals-17-00379-f003:**
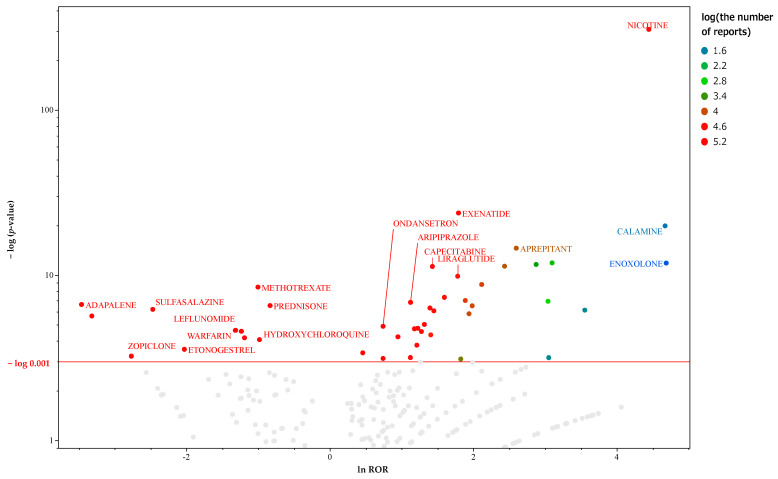
Medicines associated with hiccups in female patients. This volcano plot was created by plotting the negative logarithm of the *p*-value (−log_10_*p*) using Fisher’s exact test on the *y*-axis and the natural logarithm of the ROR (lnROR) on the *x*-axis. The red line on the *y*-axis represents *p* = 0.001. In female patients, drugs with >1000 reports and those considered significant (*p* ≤ 0.001 using Fisher’s exact test) are included in the volcano plot. The color of the individual points represents differences in the log of the number of reports for each drug. The gray plots indicate drugs with *p* > 0.001 or with <1000 reported cases. In this scatter plot, the signal is larger for the points (drugs) plotted in the upper right corner. The blue-to-red colors represent the number of times an adverse drug reaction was reported.

**Figure 4 pharmaceuticals-17-00379-f004:**
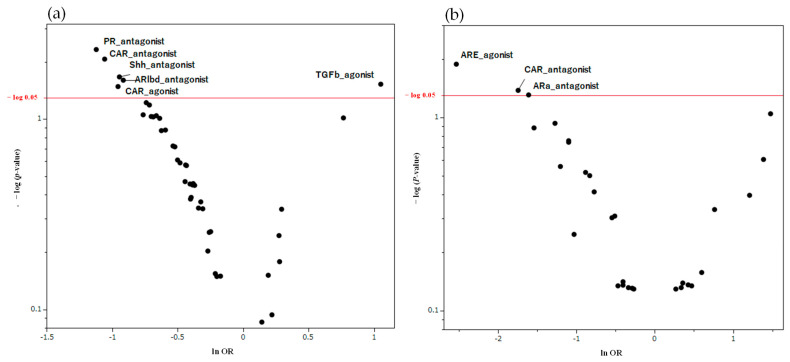
MIEs associated with hiccup induction in pharmacotherapy. This volcano plot was created by plotting the negative logarithm of the *p*-value (−log_10_*p*) using Fisher’s exact test on the *y*-axis and the natural logarithm of the OR (lnOR) on the *x*-axis. The red line on the *y*-axis represents *p* = 0.05. In this scatter plot, signals associated with hiccup induction (positive signals) are plotted in the upper right, whereas those associated with inhibition (negative signals) are plotted in the upper left. (**a**) is the volcano plot for the male table, whereas (**b**) is the volcano plot for the female group. PR, progesterone receptor; TGFb, transforming growth factor-beta; Shh, sonic hedgehog signaling; ARlbd, androgen receptor lbd; ARE, antioxidant response element; CAR, constitutive androstane receptor; ARa, androgen receptor with antagonist.

**Figure 5 pharmaceuticals-17-00379-f005:**
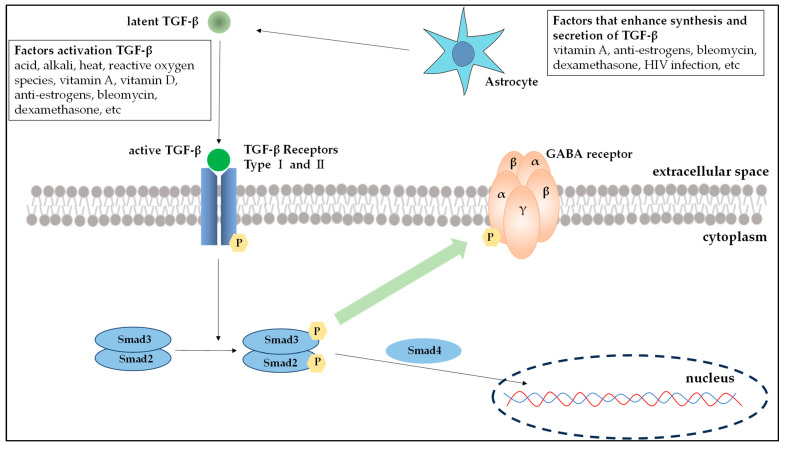
TGF-β signaling and the GABA neurotransmission interaction. The TGF-β superfamily interacts with receptors located on the plasma membrane. Upon binding with TGF-β, type 2 receptors undergo phosphorylation and subsequently form a complex with type 1 receptors. Activation of this receptor complex leads to the phosphorylation of Smad, an intracellular signaling molecule, which then forms a complex. It has been hypothesized that the interaction of TGF-β with GABA receptors may inhibit the phosphorylation of gephyrin, a scaffolding protein essential for GABA receptors, thereby enhancing GABAergic neurotransmission. However, the specific details of this interaction and its mechanisms remain significantly unexplored.

**Figure 6 pharmaceuticals-17-00379-f006:**
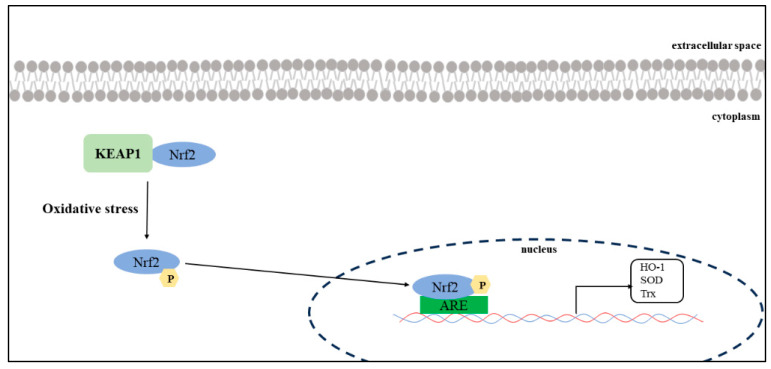
Oxidative stress and Nrf2–ARE signaling. Nuclear factor erythroid 2-related factor 2 (Nrf2)–ARE is an important signaling pathway that regulates the expression of antioxidant enzymes. Nrf2 binds to KEAP1 in the cytoplasm and maintains low activity under normal physiological conditions. If cells are exposed to oxidative stress, Nrf2 unbinds KEAP1 and translocates into the nucleus to bind with ARE. This promotes the transcription and expression of a series of antioxidant enzymes, including heme oxygenase-1 (HO-1), superoxide dismutase (SOD), and thioredoxin (Trx).

**Figure 7 pharmaceuticals-17-00379-f007:**
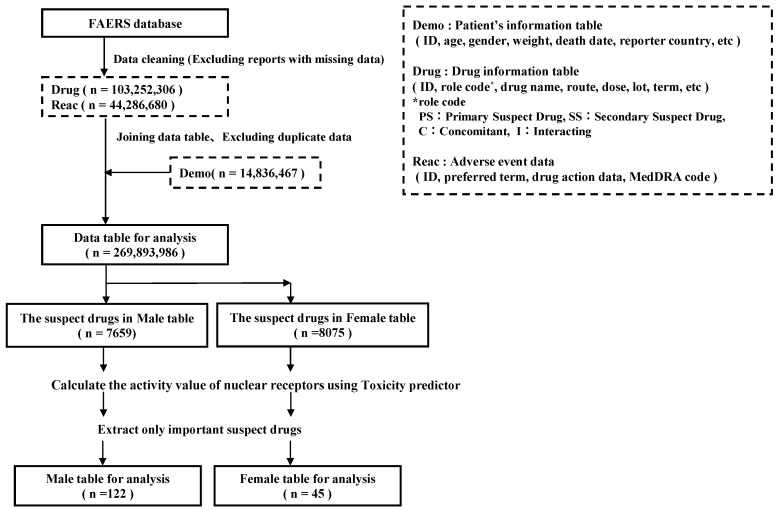
Data analysis table. The FAERS data used in this study comprised five tables: demographic characteristics of the patients and management information (DEMO), drug and biological information on reported adverse events (DRUG), and Medical Dictionary for Regulatory Activities (MedDRA) terminology coding adverse events (REAC). Duplicate data were removed from each table. The DRUG and REAC tables were incorporated into the DEMO table using the primary ID. This table was defined as the data table for analysis. All patient data contained in the DEMO table were used. Using this data table for analysis, the male and female subgroups were created.

**Table 1 pharmaceuticals-17-00379-t001:** Univariate and multivariate analyses of MIEs associated with drug-induced hiccups.

		Univariate Analysis				Multivariate Analysis				
MIEs	Activity Type		95% CI				95% CI			
		Odds Ratio	Lower	Upper	*p*-Value (Fisher’s Exact Test)	Odds Ratio	Lower	Upper	*p*-Value (Likelihood Ratio Test)	*p*-Value (Wald Test)
ERaant	agonist	0.21 *	0.06	0.69	0.010	0.28	0.06	1.26	0.088	0.097
PR	antagonist	0.43 *	0.23	0.82	0.011	0.61	0.30	1.26	0.181	0.181
ARaant	agonist	0.33 *	0.14	0.76	0.011	0.65	0.21	1.97	0.445	0.443
TGFb	agonist	3.37 *	1.30	8.75	0.014	4.59 *	1.54	13.67	0.002	0.006
PR	agonist	0.20 *	0.05	0.76	0.014	0.66	0.12	3.55	0.625	0.628
Shh	agonist	0.40 *	0.18	0.89	0.026	0.63	0.25	1.60	0.330	0.329

MIEs: molecular initiating events, ERaant: estrogen receptor alfa with antagonist, PR: progesterone receptor, ARant: androgen receptor with antagonist, TGFb: transforming growth factor-β, Shh: sonic hedgehog. *: indicates significant odds ratios in the univariate and multivariate analyses.

**Table 2 pharmaceuticals-17-00379-t002:** Univariate and multivariate analyses of MIEs associated with drug-induced hiccups in the male group.

		Univariate Analysis				Multivariate Analysis				
MIEs	Activity Type		95% CI				95% CI			
		Odds Ratio	Lower	Upper	*p*-Value (Fisher’s Exact Test)	Odds Ratio	Lower	Upper	*p*-Value (Likelihood Ratio Test)	*p*-Value (Wald Test)
PR	agonist	0.326 *	0.152	0.698	0.005	0.60	0.22	1.62	0.318	0.318
CAR	antagonist	0.347 *	0.160	0.753	0.009	0.56	0.23	1.41	0.222	0.220
Shh	agonist	0.389 *	0.176	0.861	0.021	0.70	0.25	1.97	0.504	0.502
ARlbd	antagonist	0.401 *	0.189	0.851	0.025	0.62	0.22	1.70	0.351	0.348
TGFb	agonist	2.857 *	1.100	7.423	0.030	3.67 *	1.29	10.45	0.011	0.015

PR: progesterone receptor, CAR: Constitutive androstane receptor, Shh: sonic hedgehog, ARlbd: androgen receptor lbd, TGFb: transforming growth factor-β. *: indicates significant odds ratios in the univariate and multivariate analyses.

**Table 3 pharmaceuticals-17-00379-t003:** Univariate and multivariate analyses of MIEs associated with drug-induced hiccups in the female group.

MIEs	Activity Type	Univariate Analysis				Multivariate Analysis				
			95% CI				95% CI			
		Odds Ratio	Lower	Upper	*p*-Value (Fisher’s Exact Test)	Odds Ratio	Lower	Upper	*p*-Value (Likelihood Ratio Test)	*p*-Value (Wald Test)
ARE	agonist	0.08 *	0.01	0.70	0.01	0.12 *	0.01	1.32	0.048	0.083
CAR	antagonist	0.18 *	0.03	0.94	0.04	0.50	0.07	3.62	0.483	0.489
ARant	agonist	0.20 *	0.05	0.88	0.05	0.24	0.04	1.20	0.073	0.080

ARE: antioxidant response element, CAR: constitutive androstane receptor, ARant: androgen receptor with antagonist. *: indicates significant odds ratios in the univariate and multivariate analyses.

**Table 4 pharmaceuticals-17-00379-t004:** Crosstabulation and formula for the reported odds ratios (RORs) of hiccups.

	Hiccups	Other Adverse Events
Suspected drugs	n_11_	n_12_
Other drugs	n_21_	n_22_

## Data Availability

Data are contained within the article.

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
