# Peer review of "The Association between Molecular Initiating Events and Drug-Induced Hiccups"

_pharmaceuticals, 2024, doi:10.3390/ph17030379_

Round 1

Reviewer 1 Report

Comments and Suggestions for Authors

The present manuscript evaluated the mechanisms of drug-induced hiccups identifying the associated MIEs. A very engaging investigation. However, some doubts should be elucidated.

 Please clarify this section "4.3. Extraction of Drugs Suspected of Causing Hiccups": the authors considered two groups according to sex, and in each group, cross-tabulations were performed based on two categories (the presence/absence of hiccups and the presence/absence of thesuspect drug)… Only the drugs with a p-value of < 0.001 and with a ROR  > 1 were suspected to cause hiccups. In this context, I would like to ask the meaning of the presence/absence of hiccups (?); was the adverse effect "hiccup" found in the studied database?

Why the number of drugs represented in Figure 7 does not correspond with the number of medicines in the supplementary Table 1?

Supplementary file 1 – The ATC name- classification that the authors introduced appears too confusing. For better perceptibility of supplementary table 1, I propose introducing the ATC code and the pharmacodynamic properties of each active substance. The Anatomical Therapeutic Chemical (ATC) is represented by a code assigned to a particular medicine.

The authors highlighted that This study's nuclear receptor activity values were predicted using QSAR modelling of tox21 (?)  and may differ from the activity values obtained experimentally. I propose to discuss this remark considering some essential suspected drugs.

Author Response

We thank Reviewer 1 for their valuable comments, which have helped us to improve our manuscript. Please see the attached file for details.

Reviewer 2 Report

Comments and Suggestions for Authors

Paper needs to be totally reorganized.  The method section needs to be moved into its normal position for a research paper.  It was hard to understand the results and the discussion when the method section was at the end of the paper.  It also resulted in duplication of information in both the results and discussion section that could be removed if the methods were moved to the beginning of the paper, like most research papers. 

Reference section needs to be checked for correct and completeness of the citations.  The first two references are missing the page numbers and the doi. 

When you create an abbreviation like NR for nuclear receptor (line 91) you need to use it throughout the paper for those words.  

Is there a reason that Figures 1, 2, and 3 were not combined into 1a, 1b, and 1c on the same page.  The current layout makes it hard to compare these figures and their content.  It also requires the footnote for the figure to be repeated. 

In table 1 the p-value is at the beginning of the univariate analysis and at the end in the multivariate analysis, which makes it more difficult to read and is not the standard format. The meaning of the OR for MIE is not a common concept for most practitioners; what does a low OR for the MIE mean?  Without the rare data it is not possible to validate the P-value calculations; the TGFb for univariate analysis was not statistically difference whereas the multivariate analysis was different; the width of the 95% CI is wider for the multivariate analysis.  Table 2 and 3 have the similar problems with their layout and the understanding of the concept.

Also, are you trying to build the concept that this methodology can be used for other adverse reactions, if so then why is that not clear in the abstract and the introduction. 

As stated as a limitation, the critical problem with this method is the number of reports filed with the FDA over time may vary greatly between newer drugs versus more established drugs.  This would appear to bias the calculation and lead one to believe that the new drug is worse. 

Comments on the Quality of English Language

Readability needs to be improved. 

Author Response

We thank Reviewer 2 for their valuable comments, which have helped us to improve our manuscript. Please see the attached file for details.

Reviewer 3 Report

Comments and Suggestions for Authors

1.      Is the question original and well-defined?

Yes, the question is clearly defined.

2.      What is the main question addressed by the research?

The aim of the work "Association between molecular initiating events (MIEs) in drug-induced hiccups" submitted to Pharmaceuticals was to identify nuclear receptors (NRs) and stress response pathways (SPs) associated with drug-induced hiccups using Toxicity Predictor, a machine - learning model. This study used data from the FAERS database, an adverse drug reaction database, to identify the key suspected drugs causing hiccups.

3.      Is the manuscript clear, relevant for the field and presented in a well-structured manner?

Yes, the manuscript is written quite clearly.

4.      Do you consider the topic original or relevant in the field? Does it address a specific gap in the field?

This is the first study to investigate molecular initiating events (MIEs), especially NRs/SRs, drug-induced hiccups using Toxicity Predictor and FAERS.

5.      Are the results interpreted appropriately? Are they significant?

The results presented in 6 Figures, 4 Tables and the supplementary are presented clearly and clearly. Supplementary requires more time to analyze the results thoroughly.

6.      Are the figures/tables/images/schemes appropriate? Do they properly show the data? Are they easy to interpret and understand?

Yes. The descriptions under the drawings are also clear.

7.      What does it add to the subject area compared with other published material?

As the authors presented, these are the first studies of this type. They analyzed the database to present investigate molecular initiating events (MIEs), especially NRs/SRs, drug-induced hiccups using Toxicity Predictor and FAERS (from January 2004 to March 31, 2022).

8.      Are the conclusions consistent with the evidence and arguments presented and do they address the main question posed?

Yes. The conclusions are consistent.

9.      Are the cited references mostly recent publications (within the last 5 years) and relevant? Does it include an excessive number of self-citations?

Citations are appropriately selected. Of the 30 citations, 4are from the last 5 years, including two from 2023.

Author Response

Thank you for your positive feedback on our manuscript.

In response to Comment 5, we have significantly revised the content and layout of Supplement 1 to make it easier to understand.

Round 2

Reviewer 1 Report

Comments and Suggestions for Authors

The authors adequately answered the questions posed by the reviewer. I propose some minor corrections. For instance, drugs classified as ATC should be replaced by drugs classified by the ATC system. (lines: 148, 182)

Author Response

We thank Reviewer 1 for their valuable comments, which have helped us to improve our manuscript.

[Comment 1]

The authors adequately answered the questions posed by the reviewer. I propose some minor corrections. For instance, drugs classified as ATC should be replaced by drugs classified by the ATC system. (lines: 148, 182)

[Response 1]

We thank you for your comment. As you indicated, we have made the following corrections.

lines: 115, 148, 182 : drugs classified by the ATC system

Reviewer 2 Report

Comments and Suggestions for Authors

Section numbers need to be redone.

the discussion section is #4

the method section is also #4 

The sections in the last portion of the paper all  need to be renumbered. 

Author Response

We thank Reviewer 2 for their valuable comments, which have helped us to improve our manuscript.

[Comment 1]

Section numbers need to be redone.

the discussion section is #4

the method section is also #4

The sections in the last portion of the paper all need to be renumbered.

[Response 1]

We thank you for your comment. As you indicated, we have made the following corrections. As you indicated, we have corrected the section numbers throughout the paper.